# From Infection to Death: An Overview of the Pathogenesis of Visceral Leishmaniasis

**DOI:** 10.3390/pathogens12070969

**Published:** 2023-07-24

**Authors:** Carlos H. N. Costa, Kwang-Poo Chang, Dorcas L. Costa, Francisco Valmor M. Cunha

**Affiliations:** 1Centro de Investigações em Agravos Tropicais Emergentes e Negligenciados, Instituto de Doenças Tropicais Natan Portella, Universidade Federal do Piauí, Rua Artur de Vasconcelos 151-Sul, Teresina 64002-510, PI, Brazil; dorcas.lc@gmail.com; 2Department of Microbiology/Immunology, Center for Cancer Cell Biology, Immunology & Infection, Chicago Medical School, Rosalind Franklin University, North Chicago, IL 60064, USA; kwangpoochang@gmail.com; 3Departament of Physiotherapy, Centro Universitário Uninovafapi, Rua Vitorino Orthiges Fernandes, 6123-Uruguai, Teresina 64073-505, PI, Brazil; orfeuyeuridice@gmail.com

**Keywords:** kala-azar, visceral leishmaniasis, *Leishmania infantum*, *Leishmania donovani*, pathogenesis, clinical presentation, systemic inflammation, innate immunity

## Abstract

Kala-azar, also known as visceral leishmaniasis (VL), is a disease caused by *Leishmania infantum* and *L. donovani*. Patients experience symptoms such as fever, weight loss, paleness, and enlarged liver and spleen. The disease also affects immunosuppressed individuals and has an overall mortality rate of up to 10%. This overview explores the literature on the pathogenesis of preclinical and clinical stages, including studies in vitro and in animal models, as well as complications and death. Asymptomatic infection can result in long-lasting immunity. VL develops in a minority of infected individuals when parasites overcome host defenses and multiply in tissues such as the spleen, liver, and bone marrow. Hepatosplenomegaly occurs due to hyperplasia, resulting from parasite proliferation. A systemic inflammation mediated by cytokines develops, triggering acute phase reactants from the liver. These cytokines can reach the brain, causing fever, cachexia and vomiting. Similar to sepsis, disseminated intravascular coagulation (DIC) occurs due to tissue factor overexpression. Anemia, hypergammaglobulinemia, and edema result from the acute phase response. A regulatory response and lymphocyte depletion increase the risk of bacterial superinfections, which, combined with DIC, are thought to cause death. Our understanding of VL’s pathogenesis is limited, and further research is needed to elucidate the preclinical events and clinical manifestations in humans.

## 1. Infection

Kala-azar, or visceral leishmaniasis (VL), is the second most lethal tropical and subtropical disease and seventh in the loss of disability-adjusted life years [1,2]. It is caused by the protozoa *Leishmania infantum* and *L. donovani* and is transmitted by the bite of infected sand flies. VL caused by *L. infantum* is a zoonosis involving a variety of mammals, especially the dog, distributed across Central Asia, the Middle East, the Mediterranean Basin, and South and Central America. The disease caused by *L. donovani* is restricted to humans in South Asia and East Africa [3]. VL has a protracted course of clinical development with fever, wight loss, paleness, and hepatosplenomegaly. Five to ten percent of patients die of this disease, resulting from the lethal complications of bacterial co-infections and hemorrhages. Anemia, neutropenia, thrombocytopenia, hypoalbuminemia, hyperglobulinemia, high C-reactive protein (CRP), and high erythrocyte sedimentation rate (ESR) are regularly noticed. It is a significant opportunistic infection in immunocompromised individuals, particularly those living with HIV or taking immunosuppressive drugs [4]. While transmission has been rapidly and consistently falling in recent years in South Asia [5,6], the incidence in Brazil has a worrisome trend toward a gradual increase in mortality [7].

This review serves to underscore the importance of elucidating disease mechanisms for the prospect of developing new treatments. The information provided herein is based on a literature search of original publications on VL and other reviews for general subjects. The literature search was focused on articles related to clinical disease. Publications on experimental in vitro and animal infections were consulted where information was not available in human infection, i.e., all the preclinical early events. Information of relevance is included from publications on other *Leishmania* spp. and other infectious diseases. Finally, the review is centered on immunocompetent human subjects, although a few comments are made about HIV co-infected patients.

The development of VL is more exception than rule after exposure to viscerotropic *Leishmania*. Indeed, a certain proportion of the human population in a given VL endemic area are leishmanin skin tests positive, indicating previous *Leishmania* infection, but only a small proportion of them have a history of the disease development [8,9,10,11]. Based on a population-based leishmanin skin-test study, the ratio of asymptomatic to symptomatic infection was estimated to be more than 200:1 [12].

The early preclinical stages of human infection by viscerotropic *Leishmania* have defied investigation in situ due to the elusive, long and variable lengths of the incubation period. The early events of infection and host response are available from a large body of extensive work on experimental leishmaniasis in vitro and in animal models. They are included for discussion with the caveat that they may or may not reflect human infection in the real world. Notably, infection of mice or dogs with *L. donovani* or *L. infantum* does not reproduce all the clinical symptoms and signs of human visceral leishmaniasis.

In animal models, infected sand flies have been found to inoculate infective promastigotes of various *Leishmania* spp., e.g., *L. infantum*, *L. donovani*, and *L. amazonensis* into the skin. Most of them are killed in a few minutes by complement-mediated lysis, leaving few survivors phagocytosed by neutrophils or monocytes [13]. The cutaneous species, *L. major*, encounter the phagocytes via their pathogen recognition receptors (PRR) when successfully infecting C57BL/6 mice [14,15,16]. It is not known if the infected neutrophils may present antigens directly to T cells, as reported for viral antigens in human neutrophils in vitro [17] or if they are phagocytized by neutrophils that undergo apoptosis and then are ingested by monocytes—the Trojan horse hypothesis [18]. *L. major* is also directly phagocytized by human monocytes, consistent with the leishmanial intracellular parasitism of mononuclear phagocytes known for successful infection, as demonstrated in humanized transgenic mice [19]. After phagocytosis, *Leishmania* antigens are presented to B- and T- cells, and adaptative immunity develops [20]. The interaction of *Leishmania* with dendritic cells (DC) has been reviewed [21] in the context of their significance as the antigen-presenting cells in T cell immunity [22]. *Leishmania*–DC interactions in relation to the lasting immunity after the curing of leishmaniasis is of great interest for further investigation.

The migratory pathway of *L. infantum* and *L. donovani* infection is expected to begin from the skin via the regional lymph nodes to the final destinations of spleen, bone marrow, and liver. A struggle is predicted between permissive versus protective innate and adaptative immune response to determine susceptibility or resistance during this *Leishmania* migration. The battle begins involving the armamentarium, composed of the parasite’s molecules, such as lipophosphoglycan, GP63, and others, and the host’s molecules initially confronting the parasite: Toll-like receptors (TLRs) (TLR2, TLR4, and TLR9) and other types of receptors, together with the antimicrobial hydrolases/peptides of infected macrophages, reactive oxygen species, and nitric oxide produced by these and other mononuclear innate immune cells. Disease ensues when parasite molecules subvert the host response, turning it amenable for *Leishmania* intracellular survival via several levels of molecular interactions, e.g., [A] induction of regulatory cytokine IL-10 by parasitized macrophages; [B] inhibition of Th1-acquired immune response and IFN-γ release; and [C] promotion of Th2 T cells and regulatory T cell (Tregs) clonal expansion by further secretion of IL-10, IL-4, and TGF-β regulatory cytokines. This process finally results in T cell anergy and leads to disease development. Infected neutrophils and monocytes/macrophages were observed in the patients’ blood, presumably serving to spread amastigotes to other infection sites and facilitating transmission to other hosts as female sand flies take bloodmeals [23]. The host may win the battle against the infection when macrophages that phagocytose *Leishmania* produce IL-12, thereby stimulating the expansion of proinflammatory Th1 T cell clones for IFN-γ release, gaining the ability to kill the parasites. After this point, cell-mediated immunity is expected to be firmly established to control parasite proliferation. The host–parasite struggle for viscerotropic *Leishmania* to establish parasitism in the mammal hosts has been elegantly reviewed by previous workers [24,25,26,27].

Disease development is modulated by the parasite–vector–mammalian host interactions. Genomic studies of viscerotropic versus cutanotropic *Leishmania* delineate the significance of the products of the A2 gene family in parasite visceralization. A mutation in the ras-like Rag C GTPase in the mTOR pathway of BALB7c mice may contribute to the control of *Leishmania* visceralization [28]. These observations offer clues for investigation to explain why *L. donovani* variants cause human cutaneous leishmaniasis instead of VL in Sri Lanka [29]. Finally, whole-genome sequence analyses of 109 *L. infantum* isolates found parasite sequences that are strongly associated with the phenotypes of disease severity, such as mortality and kidney failure [30]. However, observations on the early preclinical events in human infection will be needed to fully understand the important question of how the disease is initiated. This might eventually be accomplished through the use of controlled human infection models (CHIM), as performed with SARS-CoV-2 [31] notwithstanding the limitations of this approach. The development of more sophisticated human 3D organoid culture systems [32] may be useful to study infection, if vasculature can be introduced for the influx of immune cells necessary for studying infection and immunity. Another useful approach under development is the digital microfluidic sensing to monitor in real time inflammation and over-reactive immune response [33].

Evidence indicates that human hosts contribute to the development of VL. Several studies have demonstrated that the human genetic background plays a role in the fate of the established infection by modulating its development into a disease state [34,35,36,37]. There is evidence of age-dependent susceptibility to VL, as indicated by the higher proportion of disease in the youngest in a given population [9,10,38]. Patients at the extreme ends of the age groups also tend to have higher parasite burdens, presumably due to their weaker immunity [39]. Both disease incidence and parasite loads are higher in males at reproductive age than in the other demographic groups [39,40,41,42,43], leading to the proposal that male sex hormones, including testosterone and dihydrotestosterone, may play a role in fueling the infection [44]. Another important host factor influencing the development of disease is immunity to prior infection. Solid immunity is clearly generated by previous *Leishmania* infections, regardless of symptomatic or asymptomatic individuals in an immunocompetent population, as both reactivation and reinfection are rarely seen [45,46]. On the same track, acquired immunosuppression strongly influences evolution of the infection in this population from asymptomatic to symptomatic [47]. Finally, malnutrition has long been considered a risk factor for neglected tropical diseases, although its role in human VL is unclear [9,10,38].

Sand flies can also modulate the outcome of *Leishmania* infection when delivering infective promastigotes experimentally into the skin of mice with diverse species of *Leishmania*. Via the bites of sand flies, it has been shown that their saliva is rich in pharmacologically active substances, which modulate vasodilation, anticoagulation, and immune response in favor of *Leishmania* successful infection. Sand fly gut microbiota may also play a certain role [48]. The success or failure of *Leishmania* infection in mice is ultimately determined by the dose of infective parasites delivered by the vectors [49,50]. Figure 1 shows the likely evolution of VL from the sand fly bite to disease development.

## 2. Disease

Successful infection leads to unchecked proliferation of amastigotes in the mononuclear phagocytes, accounting for the disease state of VL. That is marked by systemic innate response with an increase in the plasma level of proinflammatory cytokines and the regulatory cytokine IL-10 [51,52]. Parasites must overcome host resistance in order to proliferate for the disease to ensue. The emergence of Th2 and Treg susceptible phenotype over *Leishmania*-specific acquired cellular immunity is expected, as indicated by the failure of immune T cells to proliferate and to produce IFN-γ in clinical cases [53,54,55,56,57]. Since *Leishmania* produce no toxins to account for the pathology of the disease seen, the innate host immune response is solely responsible for the clinical symptoms observed [58]. Indeed, there is a positive correlation between severity of the disease and parasite burdens [39,59]. Figure 2 shows some examples of children with various signs and symptoms of VL.

## 3. Constitutional Symptoms

Table 1 summarizes the hypothetical pathogenesis of VL. VL’s signs and symptoms overlap with those of overproducing proinflammatory cytokines, like IL-6, IL-1β, TNF-α, IL-8, and other proinflammatory cytokines as well as the regulatory cytokines, IL-10 and TGF-β. All these cytokines are produced via signal transduction triggered by *Leishmania* interaction with PRRs after infection of macrophages [60]. IL-6 is thought to have the broadest spectrum of impact due to the nature of its receptors. IL-6 interacts with both membrane-bound (mIL-6R) and soluble IL-6 receptors, and the co-receptor gp130, accounting for its pleiotropic activities [61]. The binding of IL-6 to mIL-6R and gp130 induces the synthesis of the acute phase reactants (APR) in hepatocytes [62] and also the overexpression of tissue factors in endothelial cells and monocytes, triggering the coagulation cascade [63,64,65]. The gp130 co-receptor of IL-6 is expressed on the plasma membrane of most cell types, including the circumventricular organs, the preoptic area of the hypothalamus and muscle cells [66,67]. Overproduction of the aforementioned cytokines, especially IL-6 and APR are the cornerstones for the development of the symptoms observed in VL.

The APR overexpressed by the hepatocytes include CRP (C-reactive protein), hepcidin, procalcitonin, CD14, serum amyloid A protein, fibrinogen, ferritin, and complement proteins C3 and C4 [68,69,70]. The different APRs produced vary in level. CRP is highly correlated with IL-6 [71] and can increase by a few thousand folds in VL, caused by both *L. donovani* and *L. infantum* [72,73,74,75]. Albumin decreases in level during the acute phase response, hence a negative APR, regularly notable in the sera of VL patients [58,76]. ESR (erythrocyte sedimentation rate) is almost always present at a high value in VL. This is a nonprotein APR whose level is known to change in response to plasma fibrinogen levels and plasma viscosity, hence an “indirect” APR [77,78,79]. APR rises with the increase in procoagulant and the decrease in anticoagulant proteins, fueling the already activated coagulation cascade, leading to fibrinolysis and bleeding [80].

**Fever** is one of the most notable symptoms of VL. It is mediated by the proinflammatory cytokines, called endogenous pyrogens, mostly IL-1β, TNF-α and IL-6. Endogenous pyrogens are synthesized and released after PRR (likely Toll-like receptor 2) interaction with *Leishmania* in macrophages [60,81]. After reaching the hypothalamus via the circumventricular organs that lack the blood–brain barrier, the pyrogens trigger the release of prostaglandin A2 via binding the prostaglandin E receptor in the preoptic area of the hypothalamus. This is the brain region where fever is generated in neurons within the central thermoregulatory circuitries [82,83].

**Weight loss** is also a common manifestation of VL, resulting from severe malnutrition [84,85]. Patients with VL-driven malnutrition are at higher risk of death [86,87,88,89,90]. While the mechanisms are unknown, there are at least three main possibilities to account for the weight loss, all linked to persistent systemic inflammation due to the production of proinflammatory cytokines, i.e., (a) loss of appetite as part of the illness following systemic inflammation [67,91,92]; (b) fever that sharply increases the consumption of energy [93]; and (c) acute phase response that increases muscle catabolism leading to sarcopenia, thereby making amino acids abundantly available for APR synthesis in the liver [94,95,96]. The actions of TNF-α, IL-1β, and especially IL-6 on the hypothalamus and muscles appear to play decisive roles in the development of cachexia.

**Nausea** and **emesis** are infrequently encountered symptoms of VL with multiple potential causes, including iatrogenic, toxic and infectious causes, gastrointestinal disorders, and central nervous system or psychiatric dysfunctions [97]. Signals to cause these symptoms may be transmitted from the infected viscera with the blood flow to the area postrema, one of the circumventricular organs where the blood–brain barrier is absent and also by the abdominal vagal afferents to the nucleus tractus solitarius. Stimulated neurons then send signals to a central-pattern generator, which coordinates the act of emesis and to the ventral medulla and hypothalamus, reaching the cortical brain areas [98]. Vomiting has been reported to be associated with increased mortality of VL patients in both Africa and Brazil [58,86,88,89,90,99,100,101,102,103]. In these patients, it is associated with higher concentrations of IL-6 according to uni- and multi-variate analyses [104]. The chemical mediators of nausea and vomiting in VL are unknown. These are the alarm symptoms in dengue, which is, like VL, also associated with cytokine storm, involving IL6 and other proinflammatory cytokines [105]. Whether they are the mediators of nausea and vomiting awaits further investigation.

**Anemia** is also a common clinical manifestation and a risk factor for poor outcome of VL [58,86,88,89,90,100,101,102,103,106,107,108,109,110,111,112]. It is typically due to iron deficiency, with hypochromia and microcytosis marked by a low serum iron concentration [113]. However, the mechanical destruction of red cells, neutrophils, and platelets alone in the enlarged spleen, e.g., hypersplenism, contributes significantly to anemia, independent of inflammation [114]. Moreover, although most VL patients suffer from iron-deficient anemia [113], it is rarely caused by bleeding, indicative of the participation of a different mechanism in VL. This process may involve the APR of VL, hepcidin—an iron-regulating peptide. It functions to inhibit ferroportin, which mobilizes iron from all the iron-transporting cells (enterocytes, spleen, and liver macrophages) into plasma for transport into bone marrow for hematopoiesis [115,116]. Indeed, anemia is negatively correlated with the level of IL-6, the cytokine that regulates APR, including hepcidin [104]; hepcidin mRNA is negatively correlated with hemoglobin in patients with VL [117]. In summary, while the destruction of red cells by hypersplenism is partially responsible for the anemia in VL, the APR contributes to this via the iron-depleting action of hepcidin under the control of proinflammatory cytokines, especially IL-6.

## 4. Localized Symptoms

**Hepatosplenomegaly** is one of the most important clues for diagnosing VL. It is determined by lymphoid hyperplasia, as histopathological studies have revealed that the spleen and the liver are full of parasites. There is no evidence of passive hypertensive congestion to explain splenomegaly. In spite of a marked atrophy of the splenic white pulp associated with necrosis and fibrosis of thymus-dependent areas, there is an accumulation of parasites-containing macrophages and plasma cell hyperplasia, accounting for the increase in the spleen volume. The number of parasites in the spleen and its architecture disruption is associated with the activities of proinflammatory and regulatory cytokines produced in diseased humans and dogs [118,119,120,121]. In both cases, parasite proliferation is promoted by the production of TGF-beta together with IL-10, produced locally by CD25(+)Foxp3(−) T cells and systematically by CD25(+)Foxp3(+) T cells [55,122,123]. The spleen increases in size to >10 times of its normal volume and appears to play a central role in the pathogenesis of VL. Splenectomy dramatically improves patients’ clinical conditions, although it does not cure the diseased HIV-co-infected patients [124]. Some degree of hypertrophy is also seen in the liver due to the proliferation of Kupffer cells in patients infected with either *L. donovani* or *L. infantum* [119,125].

Protracted **edema** (swelling) with or without **anasarca** (generalized edema) is a consequence of APR [109] and a risk factor for death in VL [58,87,88,89,90,101,102,103,107,108,109,111,126,127]. It is caused by a reduced concentration of albumin in plasma. Albumin is a negative APR and the principal protein determinant of oncotic plasma pressure. When it is low, plasma leaks to the interstitial space, leading to edema [128]. Therefore, hypoalbuminemia is VL’s primary cause of edema [68,69,70], consistent with our observation that IL-6 is elevated in edematous patients [104]. Hepatic failure is another cause of hypoalbuminemia, but this is rare in VL [76,119].

**Cough** is a common symptom and, together with the presence of pulmonary rales and dyspnea, indicates lung involvement in VL. This symptom is caused by a combination of alveolar bacterial pneumonia and interstitial proinflammatory response due to systemic *Leishmania* infection. These symptoms deserve prompt attention, since they are significantly associated with patients’ death, as found in many independent studies [58,89,101,102,103,109,110,111,126,129]. Interstitial pneumonitis is of inflammatory nature, as suggested by histopathologic studies that have demonstrated interstitial lung involvement where bacterial was absent; amastigotes were scarce, but Leishmania antigens were detected [130]. This conclusion is reinforced by an immunohistochemistry study of autopsy samples, showing interstitial accumulation of macrophages [131] and by computer tomography findings suggestive of association with respiratory symptoms [132]. The immunohistochemistry study also revealed a pattern of regulatory Th2 cytokines, indicative of an increased likelihood of bacterial superinfections. However, the histopathological and immunological data of VL suggest that the interstitial pulmonary involvement may be included in the category of profibrotic and proinflammatory interstitial lung diseases rather than bacterial infectious diseases [133,134,135].

**Renal syndromes** and low-grade **renal failure** were frequently observed in VL, and renal failure has been associated with increased mortality, as reported in some studies [89,109,127,136,137]. **Proteinuria** also reflects disease severity [58]. However, parasites are seldom seen in the kidneys. Renal involvement is related to interstitial nephritis and distinct glomerular participation, like collapsing segmental and focal glomerular sclerosis, necrotizing segmental and focal glomerular sclerosis, and membranoproliferative lesion. The renal pathogenesis is not well understood and is likely multicausal. Due to the presence of polyclonal gamma globulins and IL-6 at high levels, it was hypothesized that proteinuria leads to proximal tubular injury, associated with glomerular inflammation [138,139]. AA amyloid glomerular deposits without mesangial hyperplasia have also been registered, likely as part of the APR [140]. However, the first suspected cause of interstitial nephritis is drug toxicity in VL, since most antileishmanial drugs are nephrotoxic [137].

**Hepatitis** is not uncommon in VL. The liver enzymes, alanine aminotransferase (ALT) and aspartate aminotransferase (AST) are slightly or moderately elevated, indicative of hepatocyte damage and useful as markers for disease severity [58,87,89,90,99,102,106,107,108,109,110,111,126,141,142]. Jaundice or an increased level of bilirubin is one of the most significant risk factors for death, particularly in patients with hemorrhages [12,58,87,89,90,99,101,102,103,107,108,109,110,111,127,129]. Many histopathology studies have demonstrated liver involvement, with ballooning degeneration of hepatocytes, Kupffer cells parasitized with amastigotes, chronic mononuclear infiltrate in the portal space, and fibrosis associated with Ito’s cells transformation into fibroblasts [119,125]. The association between liver inflammation with the elevation of IL-1β, TNF-α and IL-6 has been known in viral hepatitis, suggestive of a similar pathway to explain the liver involvement in VL [62,143]. However, liver involvement usually is not severe, as demonstrated by the rarity of hepatic encephalopathy. Our observations found that IL-6 was the serum cytokine with the strongest correlation, with an elevation of AST but not with that of ALT in a multivariate linear regression analysis [104]. As ALT is more specific to liver involvement than AST, the stronger association of AST with VL severity may reveal a more generalized, multiorgan involvement [144]. Indeed, multiorgan microthrombosis caused by disseminated intravascular coagulation (DIC), as seen in liver biopsies [125], may be another explanation for the rise of AST in VL.

**Diarrhea** is a prevalent symptom of VL caused by infection with *L. infantum*, and parasites are seen in the intestinal mucosa [126,145]. It is a consistent risk factor for death [58,86,87,88,89,90,99,100,101,102,103,106,107,108,109,111,127,146]. Severe malnutrition can lead to diarrhea through several mechanisms [147], potentially contributing to diarrhea seen in malnourished patients of VL. Since there is heavy parasitism with mucosal changes in patients with diarrhea and VL, it looks very likely that gut parasitism and local secondary inflammation are the leading cause of diarrhea in VL [126,148]. Comparison of gut mucosa of VL patients with and without diarrhea may further clarify the mechanism of its pathogenesis.

**Post-kala-azar dermal leishmaniasis (PKDL)** occurs in immunocompetent patients when infected by *L. donovani*, but only in patients with AIDS when infected by *L. infantum* [149,150]. This disparity cannot be explained fully without an in-depth study on the complexity of host–parasite–vector interactions, although the genetic differences between the two species is expected to play a role. At a time of a successful VL elimination program in South Asia, PKDL deserves renewed attention, since it is a source of parasites that fuels transmission and threatens elimination efforts. The lesions of South Asia PKDL caused by *L. donovani* are of the papulonodular and polymorphic type, with high parasite loads. Indian PKDL emerges in 5–10% of the VL patients 2–3 years after chemotherapy. They are rarely cured spontaneously. The East African type is monomorphic macular lesions with low parasite loads and patchy inflammatory infiltrates. East African PKDL emerges in 50–60% of the VL patients after or during the treatment, but 85% of the cases are self-healing. It has been proposed that PKDL occurs after cure of VL with reactivation of the specific immunity, i.e., a decrease in the levels of regulatory T cells, TGF-beta, and IL-10 and an increase in those of IFN-γ, TNF-α, and IL-12. *L. donovani* persisting in the skin may be detected by reactivated immune cells, which infiltrate into the cutaneous tissue, causing dermal inflammation by the secretion of IFN-γ, giving rise to dermal manifestations [19,151]. However, this conclusion clashes with the recent report that IL-10 remains elevated in PKDL [15]. Of relevance to mention is the report that treatment of VL with liposomal amphotericin B reduces the incidence of PKDL [152]

## 5. Laboratory Changes

**Pancytopenia** (reduced blood erythrocytes, white blood cells, and platelets) is a typical feature of VL. It has been frequently observed in hypersplenism of other etiologies under many noninflammatory conditions, e.g., the presinusoidal portal hypertension caused by portal obstruction [77,114,153]. In VL patients, all cytopenia are more severe due to systemic and splenic inflammation, exacerbating the mechanical destruction of blood cells and platelets [120,121]. The essential role of inflammation is highlighted by the facts that when the inflamed spleen of relapsing VL is taken out by splenectomy [120], leukopenia and thrombocytopenia subside rapidly and completely [124], while in noninflammatory hypersplenism, the recovery is only partial [114]. As discussed above, the anemia seen in VL is also a consequence of inflammation via the APR hepcidin, leading to iron-deficiency anemia. In VL, severe anemia has bone marrow erythroid hyperplasia and dysplasia instead of hypoplasia, as revealed by comparing the bone marrow cell population with the peripheral blood counts. This observation suggests that bone marrow plays no role in the anemia, which results from mechanical destruction of red cells in the spleen due to splenomegaly and inflammation-dependent iron deficiency [77,154].

**Neutropenia** is a prominent white cell alteration in VL. Severe neutropenia (<500 neutrophils/mm^3^) alone is not a risk factor for VL, in contrast to febrile neutropenia for bacterial sepsis [58,102,103,106,107,108,109,110,111,146]. Neutrophils are produced from myeloid lineage progenitor cells in bone marrow, which mature into nondividing cells and enter a postmitotic pool for distribution. Mature neutrophils are present as a free-flowing blood pool in the vasculature and as the marginated pool, residing in other tissues/organs, e.g., spleen, liver, and bone marrow. Up to 30% of the total mature neutrophils normally reside in human spleen, which increases proportionally with the degrees of splenomegaly, explaining the peripheral neutropenia in hypersplenism. The degree of neutropenia indeed varies with the spleen size in rats and humans with hypersplenism [114,155]. Marginated pools of neutrophils (spleen, liver, and bone marrow) may increase by inflammatory parasitization of these organs, diminishing peripheral neutrophils in conjunction with their necroptosis, pyroptosis, and neutrophil extracellular traps [156,157]. In VL, the myeloid lineage can be hypo- or hyperplastic, independent of peripheral blood neutrophil counts [77,154]. This set of observations indicates that the neutropenia seen in VL is not determined by bone marrow changes but by hypersplenism and systemic inflammation. This is consistent with the observation that *L. donovani* infection of mice increases their granulocyte colony-stimulating factor (GSF) and granulocyte-macrophage colony-stimulating factor (GM-CSF) [158]. The dynamics of neutrophil changes in the spleen, bone marrow and liver needs detailed investigation in VL patients.

**Thrombocytopenia** is a common feature of VL but is rarely low to the level that requires platelet transfusion [106,159,160]. This is considered as one of the risk factors for poor prognosis [58,87,89,101,102,103,107,109,110,111,127,129]. Thrombocytopenia may result from impaired production, increased destruction, and/or dilution of platelets [161]. Platelets emerge from the fragmentation of the cytoplasm of megakaryocytes in the bone marrow and circulate in the blood as anucleate particles. There are two main mechanisms associated with thrombocytopenia in VL: dilution of peripheral platelets due to hypersplenism and further depletion by consumption coagulopathy. There is no evidence for a reduced production of platelets in VL patients, judging from their bone marrow megakaryocyte cellular integrity [77,154]. The recovery of platelet count is much faster than the spleen size reduction in convalescent patients, suggesting that the primary cause of thrombocytopenia is associated with inflammation leading to DIC, followed by consumption coagulopathy in VL [162].

**Hyperglobulinemia**, more precisely, **hypergammaglobulinemia**, is mediated by proinflammatory cytokines in VL patients [109,163]. The hypergammaglobulinemia is polyclonal, resulting from IL-6-mediated differentiation and expansion of B-cells into antibody-producing plasma cells [61,164], producing electrophoretically heterogeneous proteins, chiefly the gamma globulin bands [165]. The numbers of these plasma cells increase in the bone marrow, gut, lymph nodes, spleen, and liver.

**Hypoalbuminemia** has been found as a risk factor of VL [58,86,87,88,99,100,101,102,103,106,107,108,109,111,126,127,129,166,167]. Hypoalbuminemia is not directly related to hyperglobulinemia because, while B-cells produce gammaglobulins, albumin is synthesized by the liver cells. However, both are initially linked to the innate response driven by IL-6 [70].

**CRP** and **ESR** are components of the APR, as stated before. CRP is a pentameric protein synthesized and secreted by the liver, which is elevated in VL. Patients with normal CRP are unlikely candidates of VL diagnosis, requiring further considerations for other diseases. CRP has no prognostic value for the disease [72,74,75,168]. ESR, readily detectable by a cheap and simple test, is also regularly elevated, and when it has low speed may indicate a poor prognosis for patients with VL [112,169,170].

**Haemophagocytic lymphohistiocytosis syndrome (HLS)** is an inflammatory hematological condition that overlaps with VL presentation in many aspects. The root cause of HLS can be genetic, iatrogenic, inflammatory, or infectious. Virally induced HLS is thought to result from the IFN-γ amplified loop of sustained antigen presentation to CD8+ T cells for activation. CRP, ESR, and **ferritin**, another APR, are elevated, responsible for the main manifestations of fever, pancytopenia, liver involvement, and marked inflammation [171]. **High triglycerides** has also been described in VL, particularly in the more severe disease [172,173]. Phagocytosis of erythrocytes, white cells, platelets, and their precursors gave the name to the syndrome but are not all diagnostic requirements. The HLS syndrome of VL does not increase its severity and requires no treatment, for example, with immunosuppressive agents, which may risk the life of patients with HLS secondary to VL [174,175].

**Amyloidosis** has been rarely described in human patients with VL, except in a few cases of HIV co-infected patients. These patients develop renal failure due to amyloidosis with elevated levels of serum amyloid. Serum amyloid is one of the positive APRs. Amyloidosis also has been described in dogs and hamsters with VL [140,176,177].

## 6. Complications

**Hemorrhage** is a frequent clinical manifestation often with the severe consequence of death in VL. The severity varies from minor signs and symptoms, e.g., epistaxis, petechiae, and bruising to hemorrhage from the gums and urinary bleeding to open digestive hemorrhage and internal bleeding, followed by hypovolemic shock [58,87,89,90,99,100,101,102,103,107,108,109,110,111,112,127,146,159]. The underlying mechanisms of bleeding appears to be associated with DIC [104,159,160] plus a set of other anticoagulant changes. DIC is characterized by systemic activation of blood coagulation with production and deposition of fibrin that is followed by events leading to microvascular thrombi in various organs, hemorrhage or both, depending on the balance of coagulation cascade. Fibrinolysis and consumption of clotting factors (including platelets) can result in hemorrhage, which predominates over the thromboembolic phenomenon [178,179,180,181,182]. Overt and non-overt DIC are almost always present in VL, as noticed in the blood test by prolonged time of coagulation, high D-dimer, and low platelet counts [104]. DIC is triggered and fueled by the presence of proinflammatory cytokines, e.g., IL-6 and IL-1β, which increase the expression of tissue factor by the endothelium and mononuclear cells to elicit the extrinsic coagulation pathway [64,80,183].

Further evidence for a role of DIC is the presence of microthrombi in the liver of VL patients [125]. DIC is potentially driven indirectly by another mechanism via the overproduction of coagulation proteins by the liver secondary to the IL-6 driven APR [80]. A slight reduction in liver function may increase the chance of bleeding when DIC is already present [184], as noted with a modest rise in the blood level of liver enzymes, i.e., ALT and AST in VL. The retention of bilirubin in this disease may also reduce the availability of vitamin-K-dependent coagulation factors [185]. Finally, hypersplenism-dependent development of milder thrombocytopenia has the potential to predispose the patients to hemorrhage when other mechanisms of fibrinolysis have already set in.

**Bacterial co-infection** is the most frequent cause of death and thus one of the riskiest manifestations in VL [58,86,87,89,101,107,108,109,146]. The co-infection can be caused by either Gram-positive or Gram-negative bacteria [169,186,187]. The coexistence of tuberculosis with VL has been reported in some parts of the world, especially in the cases of HIV co-infection [47,88,90,101,102]. Acute bacterial nosocomial infection is another risk factor due to the prolonged in-hospital stay of VL patients [186]. Their propensity to secondary bacterial infection is far from having a clear-cut explanation. Some possibilities raised include lymphocyte anergy following exaggerated regulatory cytokine response, spleen disorganization, and malnutrition. The cause-and-effect relationship has not been well-established in the case of HIV co-infection as a risk factor for susceptibility of patients to severe secondary bacterial infection in VL. We have previously demonstrated that clinically, severe VL has strong similarities with bacterial sepsis in having a wave of proinflammatory cytokines leading to severe complications and mortality, albeit the systemic inflammatory response being slow-progressing or protracted [57]. We refer to this as leishmanial sepsis [188,189,190], lacking the acute components of bacterial sepsis, like time-dependent hypovolemic shock and the solid regulatory component. The latter renders patients of bacterial sepsis unable to contain the active infection and predisposes them to new bacterial, viral, or fungal infections [191,192,193,194]. Figure 3 depicts the progression of VL towards lethality.

VL is indeed similar to bacterial sepsis in their proinflammatory and regulatory cytokine profiles. These cytokines are elevated in VL [51] and reach an extremely high level in severe disease [104,120]. One critical point of bacterial sepsis is the development of the so-called counter-inflammatory response syndrome (CARS), in which regulatory cells, cytokines (e.g., IL-10 and TGF-β), and other soluble factors inhibit the action of IFN-γ to mediate the mechanism of killing intracellular pathogens. This process is also called lymphocyte anergy due to the apoptosis of proinflammatory immune cells [191,193,194,195]. Similarly, elaboration of IL-10 and TGF-β is associated with an increased parasite burdens seen in patients with VL. Thus, CARS also occurs in VL, as a possible factor predisposing patients to the complication of bacterial infection and death.

Another potential factor predisposing patients to bacterial superinfection is lymphopenia. Indeed, some patients with bacterial sepsis suffer from profound lymphopenia, a marker of poor prognosis [196,197,198]. It has been proposed that apoptosis of lymphocytes plays a significant pathogenetic role in septic lymphopenia [199]. Moreover, patients with sepsis who received the human recombinant growth factor for the lymphoid lineage IL-7 CYT107 recovered from lymphopenia [200]. Lymphopenia was accounted for by a reduced thymic output of lymphocytes due to their apoptosis in VL-HIV co-infection [201,202]. Apoptosis of immune cells has been long described to play a role in human and animal infection and immunity [203,204]. To our knowledge, lymphopenia was proposed only in a single study as a risk factor for the death of patients with VL. Reanalysis of our previously published data [109] (Table 2) revealed that the mean and median lymphocyte counts were significantly lower in patients who died of VL than those who survived. Thus, lymphocyte depletion appears to increase the chance of secondary bacterial infections in VL as a significant cause of death. This finding raises the consideration of adjuvant therapy with recombinant IL-7 or check-point inhibitors for treating patients of VL with severe lymphopenia. While neutropenia did not contribute to the death of patients in our cases (Table 2), extreme neutropenia was noted to increase the risk of bacterial infections in other patients with the disease [58].

**Malnutrition** is another factor which has been considered to predispose VL patients to bacterial infections. Undernutrition has been suggested to be a risk factor for developing VL (primary malnutrition), although it is also known as a consequence of VL (secondary malnutrition). Prospective cohorts of children exposed to *L. infantum* have been evaluated for the effect of nutrition with equivocal outcome of whether malnutrition predisposes patients to the infection or to the development of disease [9,10]. Malnutrition has been well described and generally accepted as a risk factor for many other infections, e.g., infectious diarrhea and pneumonia [147,193]. The work in this area is inherently difficult in the attribution of a multitude of immunological defects found in primary or secondary malnutrition [205]. Malnutrition secondary to VL, as discussed earlier, is similar to that of bacterial sepsis, resulting from systemic inflammation and APR.

Interestingly, the better nourished a patient is, the lower the parasite burdens seen in patients during the progression of VL. It is possible that secondary malnutrition might also be immunosuppressive, leaving *Leishmania* proliferation unchecked [39,59]. While how secondary malnutrition affects human immunity to the parasitism is not known, it probably differs mechanistically from the primary malnutrition. For instance, proinflammatory cytokines seen in VL are known to act on the hypothalamus to drive the loss of appetite [67,91,92]. The dramatic shift-down of liver protein synthesis also noted in VL [68] may lead to immunosuppression for the following accounts. The reduced synthesis of albumin with the increasing levels of APR theoretically favors the host defense by freeing amino acids for the synthesis of immunity-related proteins, e.g., CRP and complement components. However, a deficiency in albumin reduces its important function as a carrier for the transport of micronutrients, like zinc. Likewise, other negative APR, such as transthyretin and retinol-binding protein, may reduce the availability of vitamin A for its essential function in the immune system, rendering patients with VL more susceptible to bacterial infections [70,85]. Due to the relevant role of malnutrition in immunity, its potential impact on chemotherapy is worthy of investigation.

In summary, the risk for bacterial infection in VL may include the following factors: prolonged in-hospital stay, HIV co-infection, regulatory cytokine immunosuppression, lymphopenia, T cell exhaustion, severe neutropenia, and malnutrition, the last one being the most important.

## 7. Death

It is noteworthy that among all the risk factors identified to account for the severity of VL, few are causally linked to death. Patients do not die or rarely die directly from the typical symptoms and signs of VL, i.e., fever, anemia, edema, liver failure, renal failure, hyperglobulinemia, hypoalbuminemia, vomiting, diarrhea, neutropenia, and/or thrombocytopenia. Vomiting and diarrhea may lead to fatal dehydration, hypovolemia and hydroelectrolytic imbalance, but these are uncommon or rare complications of VL. As discussed above, severe disease may lead to some degree of hepatitis and jaundice that, coupled with DIC, can mimic acute liver failure. This is, however, not the real cause of death, since typical fulminant hepatitis has not been observed in autopsies of patients with VL [119,125,141]. Acute and severe bacterial co-infection is considered to aggravate leishmanial sepsis, accelerating progression to bleeding. Therefore, inflammatory end-stage leishmanial and bacterial sepsis seem to be the main causes of death in VL, resulting from septic shock, disseminated microthrombi, and multiple organ dysfunction syndrome.

The central determinant of leishmanial sepsis in human VL proposed is exacerbated inflammation, followed by counter-inflammation and immunosuppression. Many questions regarding this proposal await further investigation. Is it primarily dependent on the human host’s variabilities, such as age, sex, HIV co-infection, and genetic background? Or are there parasite factors that trigger systemic inflammation? A high probability of 83% for an association of VL mortality with parasite genomes was suggested from analysis of the sequence data from 109 *L. infantum* isolates in conjunction with the clinical data [30]. Currently, we are expanding the database to provide statistical power with the hope to precisely identify *Leishmania* virulence factors in play for designing better pharmaceuticals and vaccines.

## 8. Concluding Remarks—Future Investigation and Clinical Managements

Early infection in human VL (and leishmaniasis in general) is a blind spot in our knowledge from the moment of the infective transmission by sand fly bite to the time when patients seek medical attention. The very early symptoms, e.g., loss of appetite and apathy, are too vague for the patients to recall the time of their onset. Fever is thus usually taken as the first sign of this disease for the ease of its measurement. Ill-defined onset of VL generates bias in longitudinal studies for assessing risk factors, such as malnutrition. To minimize this and other uncertainties requires the information for the timeframe of transition from the very initial infection to the development of human disease. Such knowledge is expected to inform us about the equilibrium between susceptibility and resistance as well as how human immunity develops.

New approaches have been proposed for the clinical management of VL based on the available knowledge of its pathogenesis presented in Table 3. Supportive therapies and assessments of biomarkers are included with the aim to improve the outcome of severely ill patients. Since IL-6 triggers DIC and the acute phase response central to the pathogenesis of VL, anti-IL-6 and its receptor antagonists are worthy of consideration to use for treating patients with cytokine storm. Evaluation of efficacy is needed after short-term and longer-term treatments. Recombinant antithrombin concentrates and recombinant thrombomodulin have been used to treat bacterial sepsis for DIC of the suppressed fibrinolytic type. These drugs may be used for VL with care due to bleeding complications. Also worthy of consideration for proof of concept is to treat VL with tranexamic acid, an antifibrinolytic drug, which has been widely used for several different clinical conditions, including bacterial sepsis. Vitamin K must be evaluated for treating VL patients with jaundice and hemorrhage. Tests for DIC polarization are recommended for investigation to optimize anti-DIC therapies in VL. It is feasible to assess IL-10 and TGF-beta as potential risk factors and as targets for neutralization to mitigate bacterial sepsis associated with VL. VL patients with absolute lymphopenia offer an excellent opportunity for testing recombinant IL-7 and immune checkpoint inhibitors to reverse lymphocyte depletion. Finally, the early use of antibiotics needs to be assessed by clinical trials, particularly when specific biomarkers are detected, indicative of bacterial sepsis [206].

How noninfected organs, like lungs, liver, and kidney are involved in the pathology of VL is largely unknown. The inflammatory trigger of DIC as a cause remains speculative. Further investigation of the infected organs, e.g., bone marrow, spleen, and liver is also desirable for understanding their susceptibility to the infection and their relative roles in host defense. The slow progression of cytokine storm in VL is of special interest for detailed investigation, as the outcome may shed light on similar events in other infectious diseases with systemic inflammation profiles, like bacterial sepsis, COVID-19, and particularly bacterial superinfection and multiple organ dysfunction syndrome. Of particular relevance for such investigation is the use of CHIM, which is apparently underway to assess vaccines of different formats against *L. major* and *L. donovani* infection challenges [26,207]. Other emerging tools of value include 3D organoid culture involving the use of human cells with different lineages [208] like immune cells, endothelium, hepatocytes, and neurons. All these new approaches are expected to become available to provide means to deal with severe disease more effectively, raising our hope for a reversal of the trend toward increasing mortality as seen in urban VL in Brazil.

## Figures and Tables

**Figure 1 pathogens-12-00969-f001:**
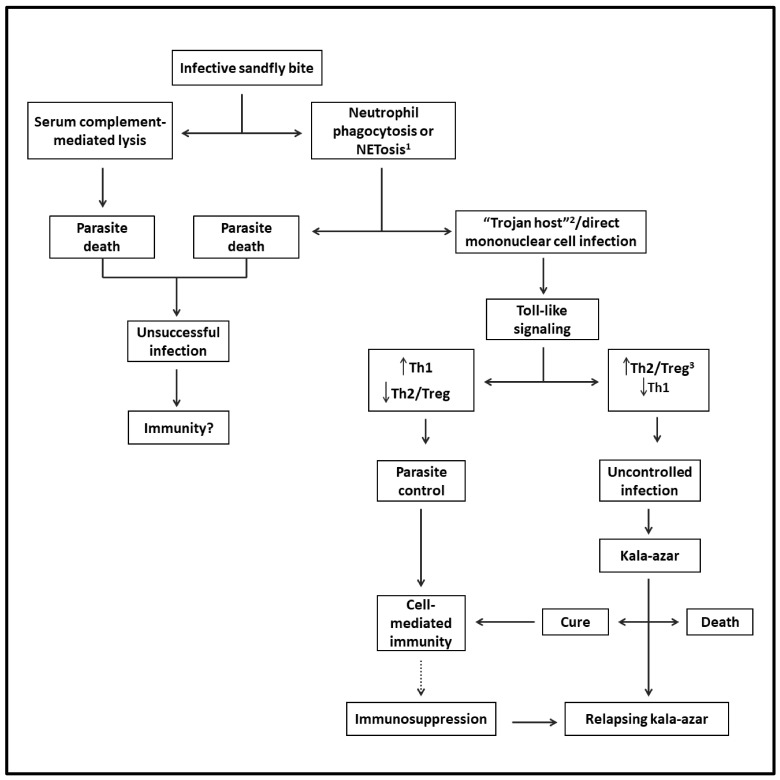
Flowchart depicting the likely evolution of VL from infection to disease: After the infective sand fly bite, parasites are released into the skin. Some perish and thus are not expected to generate a cellular immune response. Viable parasites reach their host cells directly via phagocytosis by mononuclear phagocytes and/or via phagocytosis by neutrophils (Trojan host), followed by engulfment by mononuclear cells. After triggering Toll-like intracellular signaling in either case, the infection is controlled by Th1- and cell-mediated immunity. However, the parasites may overcome this host response to replicate by mediating the generation of Th2 and Treg responses, progressing to VL. The patients may die or be cured, leading to lasting immunity to reinfection. Some patients may suffer from relapsing VL due to suppression of cell-mediated immunity preventing a complete cure. ^1^ Regulated cell death with the formation of neutrophil extracellular traps (NET). ^2^ A mechanism of intracellular cell entry through the phagocytosis of neutrophils with engulfed microbe by mononuclear phagocytes. ^3^ T-regulatory cells.

**Figure 2 pathogens-12-00969-f002:**
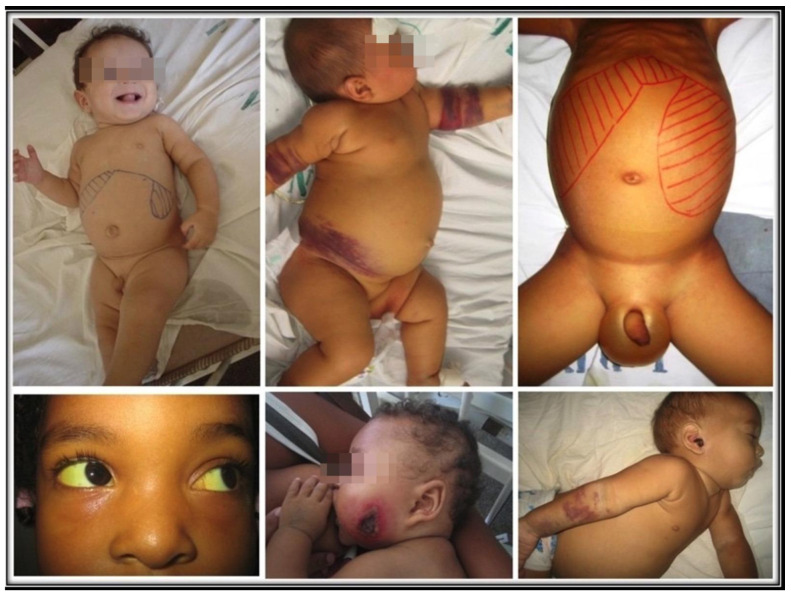
Uncomplicated, complicated, and lethal VL. Top left: an uncomplicated disease with hepatosplenomegaly and paleness. Top center: extensive bruising. Top right: large hepatosplenomegaly, with ascites and edema of the scrotum. Bottom left: scleral jaundice. Bottom center and right: *Pseudomonas aeruginosa* secondary infection in the face and the ear.

**Figure 3 pathogens-12-00969-f003:**
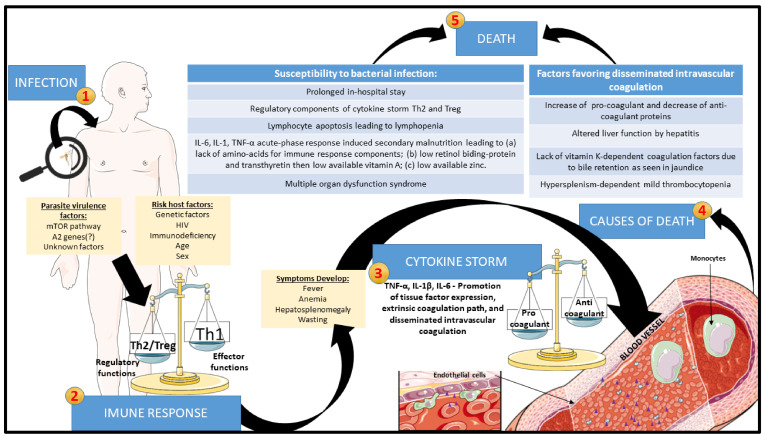
Proposed evolution of VL to the death of the patients through leishmanial sepsis: After *Leishmania* infection, the progression of the disease can be exacerbated by interactions between parasite virulence and host factors. With the development of clinical signs and symptoms, leishmanial sepsis may ensue due to exaggerated innate immune response, which promotes increased tissue-factor expression by endothelium and monocytes, generating disseminated intravascular coagulation (DIC). At the end stage, DIC can trigger bleeding enhanced by acute phase reactions with elevation of procoagulant proteins and reduction in anticoagulant proteins, with liver function and bile retention being altered, reducing vitamin-K-dependent coagulation factors and thrombocytopenia secondary to hypersplenism. Simultaneously, prolonged hospital stays of the patients in conjunction with their lymphopenia and cachexia due to the production of IL-6, IL-1, and TNF-α and deprivation of amino acids, retinol, and zinc increase their susceptibility to bacterial infection and superimposed bacterial sepsis. These episodes of leishmanial sepsis fuel multiple organ dysfunction syndrome and death of the patients.

**Table 1 pathogens-12-00969-t001:** Pathogenic mechanisms proposed to account for the clinical manifestations of VL deduced from organ/tissues and potential mediators involved in human patients and animal models.

Clinical and Laboratory Manifestation	Model	Proposed Mediator	Organ and System Involved	Proposed Mechanism
Constitutional symptom				
Fever	Rodents (rabbit, mouse, rat, guinea pig)	IL-1β, IL-6, TNF-α	Circumventricular organs, hypothalamus, preoptic area, the central thermoregulatory circuitries	Exogenous and endogenous pyrogens cross the brain barrier and reach the hypothalamus, where prostaglandin A2 acts via the prostaglandin E receptor 3, generating fever
Weight loss	Rodents (mouse, rat)	IL-1β, IL-6, TNF-α, other cytokines	Circumventricular organs, hypothalamus, muscles, adipose tissue	“Sickness behavior” with loss of appetite, plus fever increasing the consumption of energy, sarcopenia due to IL-6, and acute phase response
Nausea and vomiting	Dog, cat, ferret, nonhuman primates, human	Unknown	Circumventricular organs, central-pattern generator, nucleus tractus solitarius.	Stimuli from the viscera transported by the blood arrive in the area postrema and by the abdominal vagal afferents to the nucleus tractus solitarius. Neurons project to a central-pattern generator, which coordinates the act of emesis, and project to the ventral medulla and hypothalamus, from which cortical brain areas are reached
Anemia	Mouse, dog, human	IL-6, IL-1β, hepcidin, ferroportin. Enlarged spleen	Bone marrow, liver, enterocytes, splenic macrophages.	IL-6 and Il-1b trigger the acute phase response protein hepcidin that inhibits the iron-exporter ferroportin, depriving bone marrow from iron. Also, phagocytosis by macrophages of the enlarged spleen
Localized symptom				
Hepatosplenomegaly	Dog, mouse, hamster, human	Hyperplasia, hypertrophy.	Spleen, liver	Proliferation of macrophages and amastigotes in spleen and liver
Edema	Dog, hamster, nonhuman primates, human	IL-6, IL-1β	Liver, blood vessels	Acute phase response diminishes the synthesis of albumin. Hypoalbuminemia decreases the oncotic pressure and plasma leaks to the interstitial space
Cough and dyspnea	Hamster, dog, cat, human	IL-6, IL-13, IL-4	Pneumonitis in the alveolar interstitial space. Alveolar space	Interstitial inflammation and thickening alveolar space or bacterial pneumonia due to regulatory cytokines
Kidney failure	Hamster, mouse, dog, cat, human	Unknown, likely multicausal. Amyloid. Medication	Glomeruli, tubules	Polyclonal hypergammaglobulinemia secondary to IL-6 secretion, proteinuria leading to proximal tubular injury, associated with glomerular inflammation. Amyloid deposition. Drug toxicity
Liver involvement	Hamster, mouse, dog, human	Undetermined cytokines	Hepatocytes, bile ducts. Hepatic venules	Hepatocyte degeneration and death. Kupffer cells parasitized with amastigotes. Disseminated intravascular coagulation causing microthrombi. Fibrosis associated with Ito’s cells transformation into fibroblasts
Diarrhea	Hamster, mouse, dog, human	Parasitism. Malnutrition	Intestinal mucosa	Heavy parasitism with mucosal inflammation
Post-kala-azar dermal leishmaniasis	Human	IFN-γ, TNF-α, IL-12, IL-10	Skin	Skin immunity is reactivated after cure, leading to a decrease in regulatory T cells, TGF-β, and IL-10 levels and an increase in IFN-γ, TNF-α, and IL-12
Laboratory changes				
Neutropenia	Mouse, dog, nonhuman primates, human	Undetermined mediators	Bone marrow, spleen, liver vasculature	Parasitized and inflamed marginated pools of spleen, liver, and bone marrow may increase the disappearance of neutrophils by mechanisms such as necroptosis, pyroptosis, and neutrophil extracellular traps
Thrombocytopenia	Dog, hamster, nonhuman primates, human	IL-6, IL-1β, tissue-factor	Vasculature	IL-6, IL-1β, and TNF-α increase the expression of tissue factor, triggering the extrinsic pathway of coagulation and depleting coagulation factors, including platelets
Hyperglobulinemia	Dog, hamster, nonhuman primates, human	IL-6	Liver, bone marrow	B-cells’ differentiation into antibody, producing plasma cells under the stimuli of IL-6
Hypoalbuminemia	Dog, hamster, nonhuman primates, human	IL-6	Liver	IL-6 induces acute phase reaction that reduces the synthesis of albumin
C-reactive protein	Dog, human	IL-6	Liver	IL-6 induces acute phase reaction that increases the synthesis of C-reactive protein
Erythrocyte sedimentation rate	Human	IL-6, fibrinogen	Liver	IL-6 induces acute phase reaction that change the balance between pro- and antisedimentation factors
Haemophagocytic lymphohistiocytosis syndrome	Human	Cytokine storm, IL-6, IFN-γ, TNF-α, IL1β, IL-10	Bone marrow, systemic	Activated CD8+ T cells and macrophages stimulate each other
Amyloidosis	Hamster, human	IL-6, serum amyloid A.	Kidneys, systemic.	IL-6 induces acute phase reaction that increases the synthesis of amyloid proteins that deposit in tissues
Complications and death				
Hemorrhage	Dog, hamster, human	IL-6, IL-1β, tissue-factor.	Vasculature	Leishmanial and bacterial sepsis: IL-6, IL-1β, and TNF-α increase the expression of tissue factor, triggering the extrinsic pathway of coagulation and consumption coagulopathy. Microthrombi can lead to multiorgan failure and death
Bacterial infections	Human	IL-10, IL-4, TGF-β.	Lungs, skin, urinary tract, blood	Regulatory cytokines, lymphopenia with lymphocyte apoptosis, following cytokine storm. Secondary malnutrition leads to several immunological defects. Bacterial sepsis further exacerbating leishmanial sepsis: disseminated intravascular coagulation causing microthrombi, which can lead to multiorgan failure, septic shock, and death

**Table 2 pathogens-12-00969-t002:** Disparity of dead versus survived patients of VL in their blood lymphocyte and neutrophil counts *.

Parameter	Died (n = 66)		Survived (n = 816)	
	Mean	Median	Mean	Median
Lymphocytes/μL	1387	^1^ 1100	1960	^1^ 1565
Neutrophils/μL	1293	^2^ 945	1331	^2^ 1065

* Original data from Costa et al., 2016 [111]; **^1^**
*p*-value < 0.001 by Mann–Whitney test and 0.002 by *t*-test; ^2^ *p*-value = 0.187 by Mann–Whitney test and 0.783 by *t*-test.

**Table 3 pathogens-12-00969-t003:** Proposed new approaches to clinical management of visceral leishmaniasis.

**Biomarkers needed for the diagnosis of the following:**
DIC ^1^ polarization (suppressed and enhanced fibrinolytic-type)
Risk of bacterial infection Early bacterial infection
**Supportive therapy with the following:**
Anti-IL-6 and anti-IL-6 receptors to alleviate cytokine storm and DIC
Tranexamic acid for non-overt DIC patients
Antithrombin concentrates for DIC patients
Recombinant thrombomodulin for DIC patients
Vitamin K for patients with jaundice and hemorrhage
Antiregulatory cytokines (IL-10 and TGF-β) to modulate cytokine storm in patients at risk of CARS ^2^
IL-7 and immune checkpoint inhibitors to replenish lymphocytes
Early antibiotic usage for suspected sepsis

^1^ Disseminated intravascular coagulation. ^2^ Compensatory anti-inflammatory response syndrome.

## Data Availability

Not applicable.

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
