# Peer review of "From Infection to Death: An Overview of the Pathogenesis of Visceral Leishmaniasis"

_pathogens, 2023, doi:10.3390/pathogens12070969_

Round 1
Reviewer 1 Report
The review is an excellent value addition to the world of Leishmania research. However, the following areas need attention.
· Abstract terminology is controversial. This needs to be extensively edited
· VL is not always opportunistic as stated.
· VL is estimated to ensue in less than 1% of the incidence of infection: Is this established?
· Wasting/cachexia: use one term consistently.
· Hepatosplenomegaly and diarrhea are dependent on local parasites facilitating bacterial super infections. Is this always the case? What is the proportion of individuals having bacterial superinfections?
· Indian Subcontinent: preferable to state as South Asia
· Line no 26: Is there evidence of lymphocyte apoptosis in VL? However in CL there is evidence of T-cell apoptosis.
· Line no 34: VL mainly caused by four species of Leishmania- L.infantum , L.donovani, L.archibaldi in old world and L.chagasi in new world.
· It is a significant opportunistic infection in the immunocompromised individuals: This is one of the co-infections and should be stated such.
· The distribution of VL has changed considerably in the last few years, as South Asia contribution has decreased considerably and needs to be highlighted. In South Asia, it is usually a disease amongst immunocompetent individuals.
· Line 120 needs reframing: Finally, whole-genome sequence analyses of L. infantum isolates found parasite sequence heavily to the severity of the disease phenotypes
· Line 23: This may be accomplished through the use of controlled human infection models (CHIM), as performed with SARS: perhaps include with a word of caution.
· Line 144: Finally, malnutrition has long been considered as a risk factor for neglected tropical diseases, although this is still unsettled in VL. Reframe the sentence.
· induction of anti-100 inflammatory cytokine IL-10; stated correctly as regulatory in Line 173
· Figure: it appears from the statement that the presence of cell mediated immunity lead to immunosuppression? Rephrase
· Please elaborate how after development of cell mediated immunity there is immune-suppression in the host? Moreover, this diagram does not depict the actual mixed Th1/Th2 immune response at initial stage in human VL.
· Line 167: T-regulatory suppressive cells: restate as T regulatory cells
· In Figure 2: Please ensure confidentiality of the individuals; eyes must be covered unless an ocular problem is being shown
· Lines 188-201: Overall, in the manuscript, the pro-inflammatory or anti-inflammatory or pleiotropic or counter regulatory status of cytokines needs to be consistently stated.
· Table 1: the title ‘main possible mediators’ needs to be reframed. In this table, it would be nice if the human data is separated from animal models, so that similarities and differences between animal models and human VL is apparent. This would be an excellent value addition to a review on VL, and would highlight that where animal models have limitations as also how much can be extrapolated from animal models. Importantly, Fever and weight loss are important symptoms of human VL in human but is not mentioned in the table.
· Line 224: change weigh to weight
· Line 228: strange sentence: Loss of appetite as a sign part of the sickness behavior….
· Line 258: Is the role of hepcidin delineated in Leishmaniasis? This needs to be clarified.
· Line no 275: One of the source of counter regulatory cytokine IL-10 is CD4+ CD25+ Foxp3+ Treg cells which is not mentioned.
· Line no 422: How does IL6 induce B-cell differentiation and antibody production in VL? Please elaborate.
· Malnutrition: and its impact on the disease progression and disease severity may be mentioned as also its potential contribution towards drug unresponsiveness may be suggested. Could the variable degree of malnutrition in African vs South Asian VL account for the variable responses observed to Miltefosine and L-AmB?
· Line 355: Indian PKDL emerges in 5-10% of the kala-azar patients 2-3 years after chemotherapy. Newer references may be inserted that suggest decreased transition of VL to PKDL following L-AmB treatment for VL.
· Line 359: It is stated that ‘PKDL happens when the specific immunity is reactivated after cure of VL leading to a decrease in regulatory T cells, TGF-, and IL-10 levels and an increase in IFN-g, TNF-, and IL-12’: The levels of IL-10 are consistently raised in South Asian PKDL and so I beg to differ with this statement and it has no corresponding reference!
· Presently, the immunopathology of PKDL is reasonably well delineated and this area needs considerable upgradation of information! The importance of PKDL in the current elimination program needs to be highlighted.
· Line no 603: Authors claimed that the main cause of the death in the VL is due to bacterial infection and inflammation; these are not the norm, as with early detection and promt treatment, this is more the exception than the rule.
My overall suggestions are (i) please trim the manuscript, by at least 30%, as it is now reading like a text book and (ii) Please reframe the sentences, as many of them are not reading very well, and dilutes the quality of a potentially good review.
The manuscript needs sentences to be rephrased using scientifically and technically correct terms.
Author Response
[Pathogens] Manuscript ID: pathogens-2423214
Title: From infection to death: an overview of the pathogenesis of visceral leishmaniasis
Answers to the Reviewer 1.
Dear Reviewer 1,
Your comments were very important to improve our paper. Sincerely, thank you very much.
Please check our answers point-by-point, below.
The entire manuscript has been thoroughly reviewed and substantially revised sentence-by-sentence for conciseness and clarity. Given below are the answers to the reviewers' specific comments & suggestions.
We made a few edits when we felt that, despite the reviewers’ comments, the text was still unclear. Additionally, we decided to use visceral leishmaniasis (VL) in most of the text instead of kala-azar for clarity to most readers.
Please note that the original title was maintained as originally “From infection to death: an overview of the pathogenesis of visceral leishmaniasis”.
Comment 1: Abstract terminology is controversial. This needs to be extensively edited.
Answer 1: The abstract has been reviewed and revised, and we hope it is acceptable now.
Comment 2: VL is not always opportunistic as stated.
Answer 2: We understand that this comment refers to the abstract. We corrected it there.
Comment 3: VL is estimated to ensue in less than 1% of the incidence of infection: Is this established?
Answer 3: Previous observations have estimated the proportion of seroconverted children who developed the disease. Badaró et al. (1986) and Evans et al. (1992) estimated that the chance of developing the disease after infection was 15% and 11%, respectively. Our estimate produced a <1% chance based on skin-test conversion in a city (where cutaneous leishmaniasis is rare) as one VL in 260 infected cases (Werneck et al, 2002). All these references are cited in the article.
Comment 4: Wasting/cachexia: use one term consistently.
Answer 4: We refer to wasting only as a symptom, as it is usually used in medical practice. Cachexia is more often used in the biological sense to denote secondary malnutrition. However, due to the difficulty in distinguishing the effects of primary or secondary malnutrition, where possible, we used the terms secondary malnutrition, primary malnutrition, or simply malnutrition and restricted cachexia to the actions of cytokines. We substituted the term wasting for “weight loss.”
Comment 5: Hepatosplenomegaly and diarrhea are dependent on local parasites facilitating bacterial super infections. Is this always the case? What is the proportion of individuals having bacterial superinfections?
Answer 5: We are sorry for this confusion. We did not intend to state that bacterial infections are secondary to local parasitism in the spleen or the gut. We intend to say that the anti-inflammatory response leads to bacterial infection. We suppose the reviewer’s conclusion might be due to the statements in the original abstract. Hopefully, revisions of the Abstract made have clarified this point.
Regarding the proportion of bacterial coinfection, one study described infection occurring in around 10%-15% of all cases, but another described infection as 40-70% for the causes of death (Queiroz et al, 2004 (NOT CITED); Costa et al, 2010 (REF #58); Costa et al, 2016 REF #111).
Comment 6: Indian Subcontinent: preferable to state as South Asia.
Answer 6: We changed it accordingly.
Comment 7: Line no 26: Is there evidence of lymphocyte apoptosis in VL? However in CL there is evidence of T-cell apoptosis.
Answer 7: Yes, there are several reports on apoptosis of lymphocytes in visceral leishmaniasis of dogs, hamsters, and humans with and without HIV coinfection (Please check references #201 and #202 for HIV coinfection). Two additional references are now included to strengthen the case:
(1) Potestio M, D'Agostino P, Romano GC, Milano S, Ferlazzo V, Aquino A, Di Bella G, Caruso R, Gambino G, Vitale G, Mansueto S, Cillari E. CD4+ CCR5+ and CD4+ CCR3+ lymphocyte subset and monocyte apoptosis in patients with acute visceral leishmaniasis. Immunology. 2004 Oct;113(2):260-8. doi: 10.1111/j.1365-2567.2004.01948.x. PMID: 15379987; PMCID: PMC1782561.
(2) de Souza TL, da Silva AVA, Pereira LOR, Figueiredo FB, Mendes Junior AAV, Menezes RC, Mendes-da-Cruz DA, Boité MC, Cupolillo E, Porrozzi R, Morgado FN. Pro-Cellular Exhaustion Markers are Associated with Splenic Microarchitecture Disorganization and Parasite Load in Dogs with Visceral Leishmaniasis. Sci Rep. 2019 Sep 10;9(1):12962. doi: 10.1038/s41598-019-49344-1. PMID: 31506501; PMCID: PMC6736856.
Comment 8: Line no 34: VL mainly caused by four species of Leishmania- L.infantum , L.donovani, L.archibaldi in old world and L.chagasi in new world.
Answer 8: Only the species L. donovani and L. infantum are recognized today. L. chagasi is now recognized as L. infantum. Please check: https://www.ncbi.nlm.nih.gov/Taxonomy/Browser/wwwtax.cgi?id=5658
Comment 9: It is a significant opportunistic infection in the immunocompromised individuals: This is one of the co-infections and should be stated such.
Answer 9: The reviewer is correct. Although we mentioned the issue in the abstract, we failed to state it in the main text. Now, it has been corrected.
Comment 10: The distribution of VL has changed considerably in the last few years, as South Asia contribution has decreased considerably and needs to be highlighted. In South Asia, it is usually a disease amongst immunocompetent individuals.
Answer 10: The recent fall in the incidence in South Asia has been highlighted. Please check: “It is a significant opportunistic infection in immunocompromised individuals, particularly those living with HIV and taking immunosuppressive drugs [4]. While transmission has been rapidly and consistently falling in recent years in India [5,6], the incidence in Brazil has a worrisome trend toward a gradual increase in mortality [7].
Comment 11: Line 120 needs reframing: Finally, whole-genome sequence analyses of L. infantum isolates found parasite heavily to the severity of the disease phenotypes
Answer 11: We thank you for the comment. We rephrased it to: “Finally, whole-genome sequence analyses of 109 L. infantum isolates found parasite sequences strongly associated with disease severity phenotypes, such as mortality and kidney failure”.
Comment 12: Line 23: This may be accomplished through the use of controlled human infection models (CHIM), as performed with SARS: perhaps include with a word of caution.
Answer 12: Again, we thank for the reviewer’s comment. We rephrased it to: “This might eventually be accomplished through the use of controlled human infection models (CHIM), as performed with SARS CoV-2 notwithstanding the limitations of this approach”.
Comment 13: Line 144: Finally, malnutrition has long been considered as a risk factor for neglected tropical diseases, although this is still unsettled in VL. Reframe the sentence.
Answer 13: We changed the sentence to: “Finally, malnutrition has long been considered a risk factor for neglected tropical diseases, although its role in human VL is unclear [9,10,38].”
Comment 14: induction of anti-100 inflammatory cytokine IL-10; stated correctly as regulatory in Line 173
Answer 14: We changed as indicated.
Comment 15: Figure: it appears from the statement that the presence of cell mediated immunity lead to immunosuppression? Rephrase
Answer 15: We rephrased the last sentence of figure 1 to: “Some patients may suffer from relapsing kala-azar due to suppression of cell-mediated immunity to prevent a complete cure.”
Comment 16: Please elaborate how after development of cell mediated immunity there is immune-suppression in the host? Moreover, this diagram does not depict the actual mixed Th1/Th2 immune response at initial stage in human VL.
Answer 16: To our knowledge, the decisive moments after the infection leading to the poles of resistance or susceptibility are not evident in the literature. We wonder if this is a yes or no clear-cut situation where every contact between an infective parasite and a host cell results in definite parasitism or definitive resistance. Host-parasite interactions could be dynamic, resulting in a mixed scenario of parasitism and resistance. Under this view, the disease results from dominant susceptibility, while immunity dominates resistance to parasite replication. Resistance following treatment of immunocompetent hosts suggests that immunity was already present during the disease and prevailed after the help of drug therapy. On the other hand, relapses suggest the opposite, e.g., parasites are suppressed by immunity but are still alive and ready to replicate as long as the immune response is no longer effective. The latter situation is reinforced by several different types of opportunistic infections whose reactivation depends on the host's immune status, like herpesvirus, cytomegalovirus, tuberculosis, cryptococcosis, histoplasmosis, toxoplasmosis, and Chagas disease.
Comment 17: Line 167: T-regulatory suppressive cells: restate as T regulatory cells
Answer 17: Corrected accordingly.
Comment 20: In Figure 2: Please ensure confidentiality of the individuals; eyes must be covered unless an ocular problem is being shown
Answer 20: We blurred the eyes with the exception of the child with jaundice.
Comment 21: Lines 188-201: Overall, in the manuscript, the pro-inflammatory or anti-inflammatory or pleiotropic or counter regulatory status of cytokines needs to be consistently stated.
Answer 21: Now we are using the terminology proinflammatory, regulatory and pleiotropic cytokines.
Comment 22: Table 1: the title ‘main possible mediators’ needs to be reframed. In this table, it would be nice if the human data is separated from animal models, so that similarities and differences between animal models and human VL is apparent. This would be an excellent value addition to a review on VL, and would highlight that where animal models have limitations as also how much can be extrapolated from animal models. Importantly, Fever and weight loss are important symptoms of human VL in human but is not mentioned in the table.
Answer 22: We agree and changed the title to “Proposed mediators”. The inclusion of the animal species under the subtitle indicates the activities were detected in a particular species and that the findings in one species are usually concordant with those from the others, thereby avoiding repetitions. Some observations in animals are not possible to be repeated in humans. Moreover, we think that highlighting the differences between species would require another extensive review due to the enormous amount of original data published. We are gathering data from mice, hamsters, dogs and humans for another publication. Finally, we recognize fever and weight loss as important symptoms and were included in the first two lines of the original table.
Comment 23: Line 224: change weigh to weight
Answer 23: Done.
Comment 24: Line 228: strange sentence: Loss of appetite as a sign part of the sickness behavior….
Answer 24: Rephrased
Comment 25: Line 258: Is the role of hepcidin delineated in Leishmaniasis? This needs to be clarified.
Answer 25: Yes, at least partially, Singh et al (2018) have shown that hepcidin mRNA is inversely correlated with hemoglobin level in patients with kala-azar. We rephrased the statement.
Comment 26: Line no 275: One of the source of counter regulatory cytokine IL-10 is CD4+ CD25+ Foxp3+ Treg cells which is not mentioned.
Answer 26: We did mention that IL-10 originated from CD25(+)Foxp3(-) T cells locally and CD25(+)Foxp3(+) T cells systemically in the original description. We doubled checked this and make sure of it.
Comment 27: Line no 422: How does IL6 induce B-cell differentiation and antibody production in VL? Please elaborate.
Answer 27: Kala-azar and other acute and chronic inflammatory diseases follow the course with high concentrations of IL-6 and polyclonal hypergammaglobulinemia. The role of IL-6 in promoting B-cell differentiation and antibody production is well known. Please check reference #62 in the following link: https://www.ncbi.nlm.nih.gov/books/NBK585137/#:~:text=Hypergammaglobulinemia%20(polyclonal%20gammopathy)'%20refers,autoimmune%20disorders%2C%20and%20some%20malignancies. We changed the wording to “The polyclonal hypergammaglobulinemia seen in kala-azar is certainly due to the well-known role of B-cells differentiation into antibody-producing plasma cells under the action of IL-6”.
Comment 28: Malnutrition: and its impact on the disease progression and disease severity may be mentioned as also its potential contribution towards drug unresponsiveness may be suggested. Could the variable degree of malnutrition in African vs South Asian VL account for the variable responses observed to Miltefosine and L-AmB?
Answer 28: We added the following comment on at the end of the paragraph: “Due to its relevant role on immunity, it would be desirable to investigate whether malnutrition changes the susceptibility to drug therapy.” This is a very complex issue that also deserves a revision on its own. As commented in the text, malnutrition is secondary to the disease itself. Hypothetically, the more severe it is the worse would be the cellular immune response to Leishmania. However, we have not read any paper suggesting that the less nourished is the patient, the less is the chance to recover with the treatment. There is an interesting recent paper on the subject, although not conclusive: Kip AE, Blesson S, Alves F, Wasunna M, Kimutai R, Menza P, Mengesha B, Beijnen JH, Hailu A, Diro E, Dorlo TPC. Low antileishmanial drug exposure in HIV-positive visceral leishmaniasis patients on antiretrovirals: an Ethiopian cohort study. J Antimicrob Chemother. 2021 Apr 13;76(5):1258-1268. doi: 10.1093/jac/dkab013. PMID: 33677546; PMCID: PMC8050768.).
Comment 29: Line 355: Indian PKDL emerges in 5-10% of the kala-azar patients 2-3 years after chemotherapy. Newer references may be inserted that suggest decreased transition of VL to PKDL following L-AmB treatment for VL.
Answer 29: Excellent suggestion, thanks! We were unaware of this observation. Now, I added the information and the reference.
Comment 30: Line 359: It is stated that ‘PKDL happens when the specific immunity is reactivated after cure of VL leading to a decrease in regulatory T cells, TGF-, and IL-10 levels and an increase in IFN-g, TNF-, and IL-12’: The levels of IL-10 are consistently raised in South Asian PKDL and so I beg to differ with this statement and it has no corresponding reference!
Answer 30: Yes, you are right. IL-10 is consistently higher in PKDL, according to the reference Jafarzadeh A, Jafarzadeh S, Sharifi I, Aminizadeh N, Nozari P, Nemati M. The importance of T cell-derived cytokines in post-kala-azar dermal leishmaniasis. Cytokine. 2021 Nov;147:155321. doi: 10.1016/j.cyto.2020.155321. Epub 2020 Oct 8. PMID: 33039255. We made a change to introduce this comment and the reference.
Comment 31: Presently, the immunopathology of PKDL is reasonably well delineated and this area needs considerable upgradation of information! The importance of PKDL in the current elimination program needs to be highlighted.
Answer 31: We added this phrase in the beginning of the paragraph: “At a time of the successful VL elimination program in South Asia, PKDL deserves renewed attention, since it is a source of parasites that fuel transmission and threaten the elimination efforts.”
Comment 32: Line no 603: Authors claimed that the main cause of the death in the VL is due to bacterial infection and inflammation; these are not the norm, as with early detection and promt treatment, this is more the exception than the rule.
Answer 32: Indeed, a lot has been achieved with the kala-azar elimination program in South India. However, there are no sign of diminished transmission in other parts of the World and the 10% global mortality is not negligible.
Comment 33: My overall suggestions are (i) please trim the manuscript, by at least 30%, as it is now reading like a text book and (ii) Please reframe the sentences, as many of them are not reading very well, and dilutes the quality of a potentially good review.
Answer 33: The review has been shortened substantially by a significant effort of sentence-by-sentence review and revision. The manuscript is now more readable. The review was design to unify the knowledge of the subject and share it with the various levels of expertise, from the bench-side to the bedside. The article covers most of the relevant development of VL.
Reviewer 2 Report
The review authored by Costa et al is an exhaustive and updated description of the different aspects in the pathogenesis of kala-azar. No major issues need to be addressed, and as such, the work deserves publication under correction of minor details described below:
1. Systematic deletion of Greek characters and number in cytokine names such as TGF-β, IL-1β, TNF- α, or interferon-gamma.
2. To the best of my knowledge, abbabbreviation for interferon is IFN rather than INF
3. Minor orthographical mistakes :
Line 61: “tion, -but only a tiny proportion”.- Hyphen should be deleted
Line 70: “with the realization that”.- “Caveat” maybe a better term than “realization”, as a asuggestion.
Line 103: “TREGS”, usually the term is written as Tregs
Line 107: “Sand-flies” must be written together as “sandflies”.
Line 258 : iron from all iron-transporting cells… iron from all the iron-transporting cells
Line 286 in kala-azar; and plasma.- Remove “and”
Line 359.- “self-heal” better “self-healing
Line 371: and the spleen hyper-inflamed environment,… and the highly inflammatory environment of the spleen
Line 401: “GMCSF” is “GM-CSF”
……….Line 434: “is a pentameric protein synthesized and secreted by the liver transiently elevated in kala-azar” is a pentameric protein synthesized and secreted by the liver, which is transiently elevated in kala-azar
Line 453: “coinfected” must say co-infection.
---------Line 551, Table 2: Lymphocytes/ L, must say Lymphocytes/mL
Line 582.- “like zinc normally transported by albumin” must say “like zinc, normally transported by albumin “
Line 749: “MBio” must say“mBio”
Line 867: “Nery C henrique”, must say Nery C Henrique
Line 878: Tishkowski K, Gupta V. Erythrocyte Sedimentation Rate.; 2022. Missing data in the reference
Line 125: Review the current status of ref 125 (in press)
Line 1102: Is it the title right? Ref 174: Brun R. Antiparasitic. Clin Microbiol Infect. 2012;18:16-17
Other comments:
Even as a side- subject I missed a more detailed description on the importance of immunity with leishmanicidal drugs respect to pathogenesis, may be some lines can be dropped for this subject.
Paragraph starting at line 74.- Line 78 “pathogen recognition receptors (PRR)”.- Usually the complement receptors are not included as PRR, despite their importance in the initial intraction promastigote-macrophage. They should be included at the text
Line 93.- The battle…. Several factors contributing to the establishment of infection were describeb, but the role of antimicrobial peptides is omitted, whereas there is a consistent body of knowledge on the impairment of the initial infection by these components of the innate immunity
Line 494: “as it is not well-established if it precedes or coincides with kala-azar”.- both situations happens. In fact, leishmaniasis may appear after previous exposure to the parasite , kept as a silent infection , and burst when the patient acquire HIV infection many years after the initial infection , and the attrition of CD4+ by HIV makes the patient more susceptible to Leishmania infection see Alvar et al 2008 Clin Microbiol Rev 21, 334
Except for several orthographic mistakes, the English is good
Author Response
[Pathogens] Manuscript ID: pathogens-2423214
Title: From infection to death: an overview of the pathogenesis of visceral leishmaniasis
Answers to the Reviewer 2.
Dear Reviewer 2,
Your comments were precious to improve our article. Sincerely, thank you very much.
Please check our answers point-by-point below.
The entire manuscript has been thoroughly reviewed and substantially revised sentence-by-sentence for conciseness and clarity. Given below are the answers to the reviewers' specific comments & suggestions.
We made a few edits when we felt that, despite the reviewers’ comments, the text was still unclear. Additionally, we decided to use visceral leishmaniasis (VL) in most of the text instead of kala-azar for clarity to most readers.
Please note that the original title was maintained as originally “From infection to death: an overview of the pathogenesis of visceral leishmaniasis”.
Comment 1: Systematic deletion of Greek characters and number in cytokine names such as TGF-β, IL-1β, TNF- α, or interferon-gamma
Answer 1: Corrected.
Comment 2: To the best of my knowledge, abbreviation for interferon is IFN rather than INF
Answer 2: Corrected.
Comment 3: Line 61: “tion, -but only a tiny proportion”.- Hyphen should be deleted
Answer 3: Corrected.
Comment 4: Line 70: “with the realization that”.- “Caveat” maybe a better term than “realization”.
Answer 4: Suggestion accepted. Thank you.
Comment 5: Line 103: “TREGS”, usually the term is written as Tregs
Answer 5: Corrected.
Comment 6: Line 107: “Sand-flies” must be written together as “sandflies”.
Answer 6: Suggestion accepted. However, I would like to paste an email from Anthony Bryceson after I provoked a discussion on the issue in the Leish-L network with the title “Sand fly or sandfly or sand-fly?”. We got many answers but this one is particularly interesting and funny: “The academics have spoken, but us lay writers could be forgiven for getting it "wrong". Usage accepts sandfly on this side of the Atlantic. As Bob pointed out the divide between sandfly and sand-fly is the Atlantic Ocean. The Oxford English Dictionary and Collins offer only sandfly, Webster offers only sand fly. Encyclopedia Britannica uses sand fly, but then includes midges and gnats. Wikipedia is utterly confused. Trawling the publications of the famous ento- parasito-logists showed that British journals such as TRSTMH use sandfly, Continental journals mostly use sandfly when publishing in English, but some such as Parassitologia seem a little more flexible and use either. US journals use sand fly, but adopted the style at different times: AJTMH in abut 1983, Science and Mem Inst Oswaldo Cruz by about 2000. I found only one sand-fly. So select your journal carefully if you have strong feelings.
Anthony Bryceson”
Comment 7: Line 258 : iron from all iron-transporting cells… iron from all the iron-transporting cells
Answer 7: Corrected.
Comment 8: Line 286 in kala-azar; and plasma.- Remove “and”
Answer 8: Corrected.
Comment 9: Line 359.- “self-heal” better “self-healing
Answer 9: Corrected.
Comment 10: Line 371: and the spleen hyper-inflamed environment,… and the highly inflammatory environment of the spleen
Answer 10: Corrected.
Comment 11: Line 401: “GMCSF” is “GM-CSF”
Answer 11: Corrected.
Comment 12: Line 434: “is a pentameric protein synthesized and secreted by the liver transiently elevated in kala-azar” is a pentameric protein synthesized and secreted by the liver, which is transiently elevated in kala-azar
Answer 12: Corrected.
Comment 13: Line 453: “coinfected” must say co-infection.
Answer 13: I confess that I am confused. The software Grammarly and Webster’s dictionary give both variants as correct. However, I corrected as suggested by the reviewer.
Comment 14: Line 551, Table 2: Lymphocytes/ L, must say Lymphocytes/mL
Answer 14: Corrected.
Comment 15: Line 582.- “like zinc normally transported by albumin” must say “like zinc, normally transported by albumin “
Answer 15: Corrected.